# Beyond TF-IDF: Reproducibility and Generalizability of Two-sided Fairness Tradeoffs

## Abstract

In this paper, we reproduce the experiments conducted by Greenwood et al. (2024) to validate their findings.[1] We successfully recreate their main empirical results and conclude that the authors' two primary claims are valid: (a) item fairness tends to impose a higher cost on user fairness in more homogeneous populations, and (b) the cost of misestimation is high and item fairness constraints do not affect this cost. We extend their experiments by using SPECTER embeddings and the Amazon Book dataset to validate the generalizability of their results. We find similar results, for both the original dataset and the Amazon Book dataset, consistent with those of the authors. However, the results obtained with SPECTER embeddings do not fully support the two claims.

## 1 Introduction

*Two-sided fairness* (or *multi-sided fairness*) aims to balance user fairness and item fairness in recommendation systems. User fairness is a measure of the relevance of the recommended articles, and item fairness is a measure of the exposure of the articles of the recommendation. Taking item fairness into account when designing a recommendation system is beneficial for users, items, and the platform itself (Wang et al., 2023). Users can access a wider range of niche information, and less popular items can reach more users. However, this is often not the case, as exemplified by the recommender system by Greenwood et al. (2024): more than 47% of papers have a probability lower than 0.0001% of being recommended to any user. Item fairness encourages providers to post more diverse content, as they know that it will not be overlooked by mainstream content, which in turn can increase user satisfaction, ultimately benefiting the platform itself (Jin et al., 2023). However, simply introducing item fairness in a recommendation system can come at a cost to user fairness. *Two-sided fairness* tries to define the balance between user and item fairness as an optimization problem (Burke et al., 2018; Chen et al., 2024; Wang & Joachims, 2021; Rahmani et al., 2022; Greenwood et al., 2024).

Greenwood et al. (2024) identified the effects of two-sided fairness on user fairness as an open question. Specifically, they were interested in finding settings in which both objectives could be simultaneously maximized. To analyze the problem of *two-sided fairness*, Greenwood et al. (2024) introduce a theoretical framework to characterize solutions for multi-sided fair recommendations (Greenwood et al., 2024). With their theoretical framework, the authors gain insights into the tradeoffs between the price of fairness and misestimation in recommendation systems. Using combined data from arXiv and Semantic Scholar (arXiv.org submitters, 2024; Kinney et al., 2023), the authors developed a recommendation engine to verify two phenomena they found in their theoretical model: (a) groups in which the users have similar preferences experience steeper user-item fairness tradeoffs than diverse groups and (b) imposing item fairness constraints to users whose preferences are uncertain will worsen their recommendations further (Greenwood et al., 2024). When testing this empirically, they found that claim (a) was supported, but claim (b) was not, as the recommendations for users with uncertain preferences were already so poor that imposing item fairness constraints did not worsen the recommendations further.

---

[1]After conducting our reproducibility study, we found multiple versions of the paper online. We based our study on the version found on arXiv, submitted on December 5, 2024. The link is provided in the references (5.3).

In this paper we aim to reproduce the authors' empirical findings, verify the validity of their claims and conduct additional experiments to assess the generalizability of their results. Specifically, we reproduce their experiments with another data set (Amazon Book data set) and with SPECTER embeddings in addition to TF-IDF embeddings.

## 2 Scope of reproducibility

Greenwood et al. (2024) developed a recommender system for arXiv that recommends papers to authors. The recommendations are based on the similarity between the researchers' previous work and potential recommendations. They tested the performance of their recommendation system and then tried to validate their theoretical findings with this system. Based on their results, the authors first conclude that fair recommendation systems should not impose item fairness constraints on *sub-markets* (sub-groups), but should instead focus on the entire set of users (Greenwood et al., 2024). If user preferences are homogeneous, item fairness constraints will naturally degrade the recommendation quality for the users. This is because if the group has utility $\epsilon$ for the first item, imposing the same utility $\epsilon$ on a second item naturally will result in some users not getting their preferred item (Greenwood et al., 2024).

Secondly, they find that fair recommendation systems should be especially mindful of users whose preferences are misestimated. Cold-start users would typically be recommended the most popular items (e.g. the average of previous recommendations). Item-fair recommendation algorithms however, will recommend them the globally least preferred items, because the preferences of these users are estimated as weaker, compared to already established users (Greenwood et al., 2024). This advice is based on their theoretical framework, but when tested, they found that the recommendations for those users were already so bad that imposing item fairness constraints did not worsen them.

In this paper, we will analyze and validate these two main findings by reproducing their empirical setup. More specifically, we will conduct the experiments done by Greenwood et al. (2024) and identify whether they support the claims listed below:

- **Claim A**: Item fairness tends to impose a higher cost on user fairness in more homogeneous populations.

- **Claim B**: The cost of misestimation is high and item fairness constraints do not affect this cost.

In section 3 we will explain the authors' model, the datasets used and their experimental setup and code. Section 4 will cover our findings, both for the reproducibility of the original study and our additional experiments. Finally, in section 5, we will summarize the extent to which their work is reproducible.

## 3 Methodology

In this section, we cover the methodology of our reproducibility report. This includes a description of the theoretical model of multi-sided fairness as developed by Greenwood et al. (2024), a description of the datasets used in the experiments, and the experimental setup. We have attempted to stay as close as possible to the original setup, as specified by the authors in their paper and GitHub repository. Their repository is publicly accessible.[2]

### 3.1 Model description

#### 3.1.1 Defining fairness

Greenwood et al. (2024) define a recommendation policy for a system with $m$ users and $n$ items as a tuple of probability distributions $\rho \in \Delta_{n-1}^m$. A recommendation for user $i$ is generated by sampling from the distribution $\rho_i$. The utility of recommending an item $j$ to a user $i$ is $w_{ij} > 0$ for both the user and the

---

[2]https://github.com/vschiniah/ArXiv_Recommendation_Research

item. The utility matrix $w$ defines how useful every recommendation of an item to every user is. The normalized expected utility for single user $i$, is normalized by $\max_j w_{ij}$, the utility a user would get from being recommended their best match. Similarly, the normalized item utility for an item $j$, is normalized by $\sum_i w_{ij}$, the items utility from being recommended to every possible user. The utilities can be defined as:

$$U_i(\rho, w) = \frac{\sum_j \rho_{ij} w_{ij}}{\max_j w_{ij}}, \qquad (1) \qquad\qquad I_j(\rho, w) = \frac{\sum_i \rho_{ij} w_{ij}}{\sum_i w_{ij}}. \qquad (2)$$

For a given policy $\rho$, user fairness is measured as the minimum normalized utility across all users and item fairness is defined as the minimum normalized utility across all items:

$$U_{\min}(\rho, w) = \min_i U_i(\rho, w), \qquad (3) \qquad\qquad I_{\min}(\rho, w) = \min_j I_j(\rho, w). \qquad (4)$$

### 3.1.2 Defining the multi-sided optimization problem

The authors define the two-sided fairness objective as maximizing user fairness subject to an item-fairness constraint. So, the optimal two-sided recommendation solution is the policy $\rho$ that maximizes the minimum normalized user utility, subject to the constraint that the minimum normalized item utility is at least a fraction $\gamma$ of the optimal item fairness level:

$$\begin{aligned} U^*_{\min}(\gamma, w) = \max_\rho U_{\min}(\rho, w), \\ \text{subject to } I_{\min}(\rho, w) \geq \gamma I^*_{\min}(w), \end{aligned} \qquad (5)$$

where $U^*_{\min}(w) := \max_\rho U_{\min}(\rho, w)$ and $I^*_{\min}(w) := \max_\rho I_{\min}(\rho, w)$. In the above formula, $\gamma$ controls the tradeoff between user and item fairness. A higher $\gamma$ means therefore that user fairness is more constrained by item fairness.

### 3.1.3 Defining the price of fairness and misestimation

The 'price of fairness' is defined as the decrease in user fairness with maximal item fairness constraint compared to the overall optimal solution. Equation 6 aligns with Claim A of the authors, highlighting the tradeoff between user and item fairness. In 6, $w$ is the utility matrix and $U*$ is the optimal user fairness given the item constraint $\gamma$.

$$\pi^F_{U|I} := \frac{U^*_{min}(w) - U^*_{min}(\gamma = 1, w)}{U^*_{min}(w)} \qquad (6)$$

The 'price of mistestimation' is defined as the decrease in user fairness with misestimated utilities compared to the user fairness with correctly estimated utilities. In equation 7, $\hat{w}$ is the misestimated utility matrix and $\hat{\rho}(\gamma)$ is a policy that solves the two-sided fairness optimization problem with misestimated utilities, that is, $\hat{\rho}(\gamma)$ attains $U^*_{min}(\gamma, \hat{w})$. Equation 7 aligns with claim B of the authors, emphasizing the impact of utility misestimation on user fairness under item fairness constraints.

$$\pi^M_U(\gamma, w, \hat{w}) := \frac{U^*_{\min}(\gamma, w) - U^*_{\min}(\hat{\rho}(\gamma), w)}{U^*_{\min}(\gamma, w)} \qquad (7)$$

### 3.1.4 Modeling choices

The authors make several choices when defining their model, which they also address in their work. The first choice is that the utility $w_{ij}$ of recommending item $j$ to user $i$ is the same for the user and the item. This means that both benefit equally from a successful recommendation. The authors thereby imitate, for instance, click-through rates, where the engagement of a user with an item is beneficial for both parties. For example, the authors want users who engage with their papers and users want papers which are useful

to them (Greenwood et al., 2024). The second choice is that user and item fairness are defined as the minimum normalized user and item utility, also known as egalitarian fairness. The authors do this to emphasize individual fairness rather than group fairness, and to focus on capturing widespread individual-level disparities without relying on group identity (Greenwood et al., 2024). We followed this definition of fairness as well. In the appendix, they also show the results for Nash fairness (product of utilities) and sum-of-$k$-min fairness (sum of utilities of $k$ worst-off users). These experiments supported Claim A as well but especially the sum-of-$k$-min fairness did not support claim B. The third choice is that in their two-sided fair optimization problem they use an item fairness constraint rather than an added term to the formula. As they indicate, the assumptions made would hold either way (Greenwood et al., 2024).

## 3.2 Datasets

### 3.2.1 ArXiv and Semantic Scholar data

The authors created the users and items of their recommendation system with data from arXiv and Semantic Scholar (arXiv.org submitters, 2024; Kinney et al., 2023). They selected the data in the category 'Computer Science' that was available on both arXiv and Semantic Scholar. They specify in their paper that they used $139,308$ papers by $178,260$ distinct authors before 2020 as the train set (users) and $14,307$ papers uploaded to arXiv in 2020 as the test set (items). When running their provided code we got different numbers for both the train and test set: $82,737$ train papers and $23,534$ test papers. We believe that this discrepancy arose due to the provided code of the authors being partially incomplete, which required us to fill in the missing parts. Details of what we changed to make the code work can be found in section 5.2. We compared how our data was distributed over the different subcategories in Computer science with how their data was distributed. Even though we found some differences, we believe that these will not affect the outcome of our experiments. The results of this comparison can be found in Appendix D.

For the validation of their recommendation engine, the authors investigated the citations of $1,128$ authors and $14,307$ papers. A recommendation of paper $j$ to user $i$ was considered good if the user cited the paper in the works that they published after 2020 or if the paper cited the author. Once again, it is not clear in the authors' code where the specific number of $1,128$ authors for the validation comes from but for the sake of reproducibility, we randomly sampled the same amount of authors. Because the authors have utilized their whole test set for the validation, we did so as well.

In Appendix C we evaluate the recommendation engine and generate Figure 1, but this time keep their original ratio of 1/10 test/train intact. This means that we discarded a big chunk of our test set, and continued with a test set size of $8,497$. We found that the difference was minimal, and therefore decided to continue with all our available data.

### 3.2.2 Amazon Books Reviews dataset

To assess the generalizability of the results obtained by Greenwood et al., we performed the same experiments with a different dataset, namely the Amazon Books Reviews dataset (Mohamed Bekheet, n.d.). The dataset consists of roughly 3 million reviews for $212,404$ unique books and $1,008,972$ different authors. First, we pre-processed the dataset by only retaining authors with 15 or more reviews and books with at least 10 reviews. Reviews with scores below four (out of five) are excluded, and we removed reviews of books lacking descriptions. This resulted in a refined dataset containing 375,870 reviews. Next, we divided the dataset into a train and a test set randomly assigning 90% and 10% of the books to the train and test set, respectively.

We validated the recommendation engine using the same metrics as described in Section 4.1.1, randomly selecting 1,000 users to perform the validation.

## 3.3 Hyperparameters

An overview of all the parameters that should be specified in order to recreate the Figures 1a and 1b can be found in Table 1. Generally, we keep all parameters the same as specified in the authors `readme` file of

their code. One difference being that we sample less authors and papers than Greenwood et al. (2024), due to computational limitations. Instead of 500 authors and 200 papers, we sample 100 authors and 40 papers.

Table 1: Overview of the hyperparameters necessary to create the figures

| Parameter | Description | Default |
|-----------|-------------|---------|
| n | Number of items | 40 |
| m | Number of users | 100 |
| beta | Proportion of the population to be correctly estimated | 0.9 |
| curves | Number of sample curves | 10 |
| curve_pts | Number of different values of gamma | 50 |
| k | Number of users to consider to calculate fairness | 1 |
| delta | Relaxation factor for total utility constraint | 1 |
| ci | Show confidence interval of curve | 1 |
| components | Number of PCA components to pre-process author embeddings for k-Means clustering | 2 |
| clusters | Number of k-Means clusters to generate homogeneous author sub-groups | 25 |

## 3.4 Experimental setup and code

The code that we used to run the experiments conducted in this paper is publicly accessible.[3]

### 3.4.1 ArXiv and Semantic Scholar data

The authors conducted a logistic regression to evaluate the predictive capability of their recommendations. The logistic regression was performed with a similarity matrix $w$. A recommendation of paper $j$ to user $i$ was considered good if the user cited the paper in the works that they published after 2020. As mentioned before, in their experiments users were authors who had published at least one paper in the training set (papers uploaded before 2020). The items to be recommended were the papers in the test set (uploaded in 2020).

**Recommendation engine**   Two variations of the engine were made: in one variation the training and test datasets were embedded using Term Frequency - Inverse Document Frequency (TF-IDF) (Rajaraman, 2011) and in the other, they were embedded with SPECTER (Cohan et al., 2020). For the TF-IDF embeddings, the authors generated a frequency vector for its abstract, which was used as embedding for the TF-IDF model. They did not publish the code they used to create SPECTER embeddings, but we followed their approach as specified in their paper. This means that we embedded the concatenation of the abstract, categories, and title of the paper using the AllenAI SPECTER model (Cohan et al., 2020). The users were not directly embedded, rather we embed each of the users authored papers and then calculate the similarity of each paper with the items.

The embeddings were then used to calculate the matrix $w$, where the utility score $w_{ij}$ for author $i$ and article $j$ was defined as the cosine similarity between the papers author $i$ uploaded and paper $j$. They made two variations: in one variation the mean of all the papers embeddings of authors $i$ was calculated and $w_{ij}$ was the similarity between that mean and paper $j$. In the other variation, $w_{ij}$ was the maximum similarity of similarities between paper $j$ and the papers $i$ wrote. So in total, they tested four variations (TF-IDF/SPECTER, *mean/max*).

**Evaluation**   They evaluated each recommendation engine using four different metrics: (a) Coefficient, (b) Std. Err, (c) Z-value and (d) Adjusted $R^2$. Based on the results, they chose to use the recommendation engine with TF-IDF embeddings and the maximum similarity for their experiments. Another reason why they chose *max* over *mean* is because *max* can capture the potential diverse interest of users. We followed

---

[3]https://anonymous.4open.science/r/reproducibility_study_user_item_fairness-EB41/

the same approach. The only difference was that our train and test sets are differently sized, as was specified in section 3.2. Moreover, we also reproduced Figure 1 with the SPECTER-embeddings.

The authors used $14,307$ papers from the test set and sampled from a pool of $20,512$ authors from articles published prior to 2020 and 2020 to generate Figure 1. This was not computationally viable for us so we down-scaled this tenfold to $1,430$ papers and $2,051$ authors. Besides this, we reproduced the Figure exactly as specified in their paper and code.

**Figure 1a**   To test the effect of homogeneity on user utility given various levels of item fairness constraints, the authors simulated a heterogeneous population and various homogeneous populations. For the heterogeneous population, they selected $m$ authors and $n$ random papers and calculated the similarity between the authors and papers via the method described above. They then computed $U^*_{min}$ for 50 values of $\gamma$ between 0 and 1. To generate homogeneous groups, they divided all the authors into 41 groups of approximately $m$ authors using the k-means algorithm. They then randomly selected 10 of these groups, and for each of the groups calculated $w$ using $n$ randomly selected papers and computed $U^*_{min}(\gamma, w)$ for the 50 values of $\gamma$. They then averaged the results of these 10 experiments (Greenwood et al., 2024) and provided the standard deviation.

**Figure 1b**   To test the effect of misestimated preference on the user utility given various levels of item fairness constraints, the authors selected 10% of authors for whom they did not calculate their utility scores but estimated them based on the average utility of the other authors. They did this 10 times for datasets of again $m$ authors and $n$ papers. They then calculated the minimum utility score for the users with and without correct preferences given 50 values of $\gamma$ (Greenwood et al., 2024).

### 3.4.2   Amazon Books Reviews dataset

We generally followed the same approach as in the previous section to reproduce Figure 1 using the Amazon Books Reviews dataset. The goal for the recommendation engine on this dataset was to predict whether a user would review a certain book highly. We again embedded the dataset using two models: TF-IDF (Rajaraman, 2011) and SPECTER (Cohan et al., 2020) to obtain a similarity matrix $w$. Like the arXiv dataset, the embeddings were created from the concatenation of the book title, description, and categories. Similarly to the arXiv and Semantic Scholar data, we do not directly create user embeddings but rather calculate the similarity between each authored paper of the user and an item. Because the authors used the maximum similarity in their main body of the paper, we focused only on *max* when reproducing the experiments with a different dataset. For the Amazon Books Reviews dataset, a recommendation of book $j$ to user $i$ was considered good if the user reviewed the book with a score greater than 4.

To generate Figure 1 with the Amazon Books Reviews dataset we followed the exact same approach as specified in the previous section 3.4.1.

### 3.5   Computational Requirements

All the experiments conducted by us could be completed on our personal devices. The bulk of the experiments were ran on 2023 Apple Macbook Pro's, using the M2 Max chip with 32GB of ram. For the average approximated run times, see Table 2. According to Apple's Product Environmental Report for the 2023 MacBook Pro, the laptop emits 327 kg of $CO_2$ over its lifetime, with an expected usage of 3–4 years (Inc., 2023). To run our experiments, we needed 2 days. Assuming an average lifespan of 3.5 years, this means we emitted approximately 0.512 kg of $CO_2$ to conduct our experiments. For comparison, the average passenger vehicle emits about 400 grams of $CO_2$ per mile (United States Environmental Protection Agency, 2025). This suggests that our experiments emitted a little more $CO_2$ than a car does in a single mile. However, this is still a significant overestimation, as the majority of a laptop's carbon footprint comes from its manufacturing rather than its usage.

Table 2: Average runtime of experiments conducted by us while reproducing the original paper and performing additional experiments

| Experiment | (Average) Approx. Runtime |
|---|---|
| Embedding data w/ TF-IDF | 5m |
| Embedding data w/ SPECTER | 21h |
| Retrieving test data w/ API | 24h |
| Running Figure 1a | 1.5h |
| Running Figure 1b | 1.5h |

## 4   Results

In this section, we will present the results of reproducing the experiments of the authors and the additional experiments we designed.

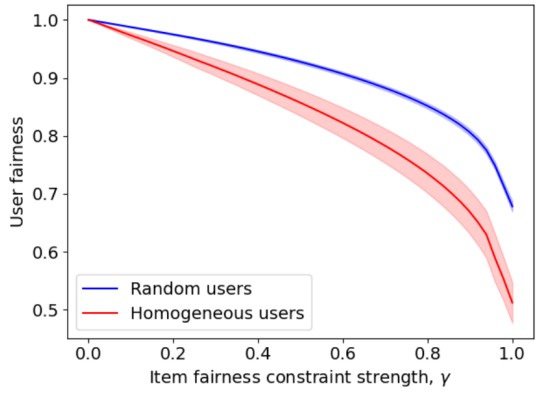

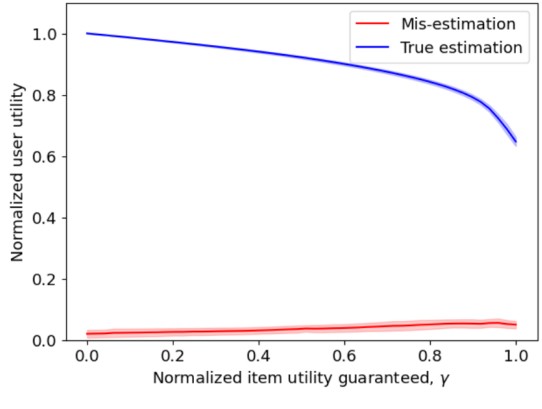

(a) Homogeneous versus diverse users.          (b) With and without misestimation.

Figure 1: Results of the experiments conducted in the paper by Greenwood et al. (2024). ArXiv and Semantic Scholar data.

The main results of the original paper were presented as Figures 1a and 1b that correspond to the two claims mentioned. Since the original paper did not provide numerical results to support these claims, we evaluated the success of our reproducibility experiments qualitatively by comparing the trends in their figures with those in ours.

### 4.1   Results reproducing original paper

#### 4.1.1   Results of logistic regression

Even though there are some subtle differences in our scores compared to the authors ones, our results follow generally the same trend as the ones of the authors, as are our adjusted $R^2$ scores very similar. The similar $R^2$ scores suggest that our model has comparable predictive performance to the one used by the authors. This means that even though we have different train and test sizes for the recommendation engine, the performance should not differ much. Consequently, we can assume that our generated Figures to support the authors' claims should follow similar trends. Greenwood et al. (2024) also provide an extended evaluation of the four tested recommendation engines in their Appendix. Because we have different train and test set sizes (section 3.2), and different distributions for these (section D), we conducted the same extended evaluation. For all methods, this includes: (a) a table of the average similarity measures, (b) density plots of the similarity scores and (c) an extensive table highlighting the logistic regression results. Our extended

evaluation can be found in Appendix A. Once again, we found that our results do not differ much compared to the ones of Greenwood et al. (2024).

Table 3: Our logistic regression results for predicting whether user $i$ cites paper $j$ from the similarity score $w_{ij}$ for each model, replication of the results by Greenwood et al. (2024). ArXiv and Semantic Scholar data

| Model | Coefficient | Std. Err | z-value | Adjusted $R^2$ |
|---|---|---|---|---|
| Max score, TF-IDF | 15.806266 | 0.140005 | 112.897568 | 0.081174 |
| Mean score, TF-IDF | 16.960261 | 0.188324 | 90.059151 | 0.050143 |
| Max score, Sentence Transformer | 21.352732 | 0.247703 | 86.203049 | 0.135880 |
| Mean score, Sentence Transformer | 18.939100 | 0.241902 | 68.292567 | 0.102387 |

### 4.1.2 Results of max score, TF-IDF model

As discussed earlier, we were not able to exactly recreate the train and test sets that the authors used for Figure 1. We therefore chose to reproduce the figure with a subset of the data. Despite this limitation, our results align with the claims made by the authors. Figure 2a demonstrates that item fairness tends to impose a higher cost on user fairness in more homogeneous populations. We can see in the figure that the homogeneous group (the red line) experiences a steeper user-item tradeoff than the diverse group (the blue line). In contrast, imposing item fairness constraints on groups with diverse preferences significantly benefits individual items with little cost to the users. As shown in the figure, user fairness for random users drops only marginally under higher item fairness constraints. This group is affected more severely only when $\gamma$ approaches its maximum value of 1.

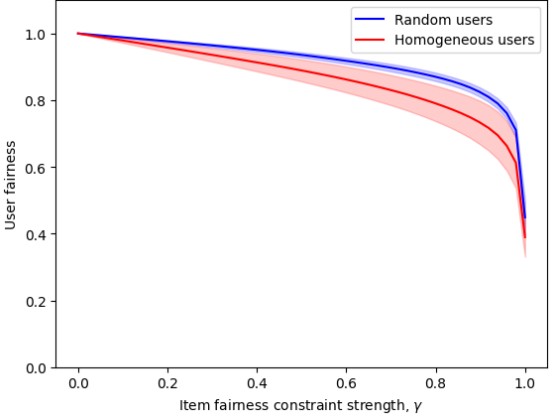
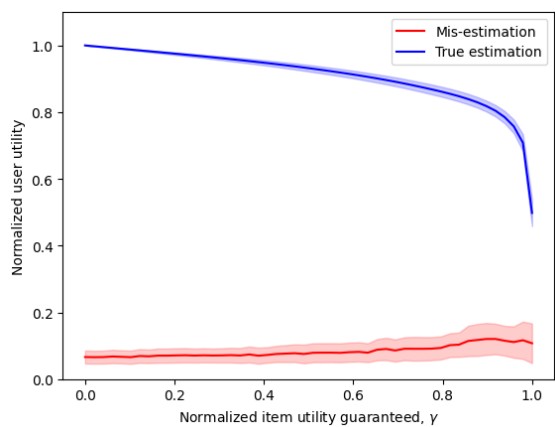

(a) Homogeneous versus diverse users.
(b) With and without misestimation.

Figure 2: Reproduction of Figure 1 of Greenwood et al. (2024), with a subset of the arXiv and Semantic Scholar data. (a) Shows the difference between random and homogeneous user groups when item fairness constraints are being imposed. Supporting Claim A, we can see homogeneous groups experience steeper user-item fairness tradeoffs compared to groups with diverse preferences. (b) Shows that the cost of misestimation for cold-start users is already so high that item fairness constraints do not affect this cost, supporting claim B.

Figure 2b illustrates that preference misestimation for cold-start users is already so high that item fairness constraints do not affect this cost. As we can see in the figure, users whose preferences are misestimated (indicated by the red line) already receive significantly less utility from their recommendations. When item utility is increased by setting $\gamma$ higher, this user utility does not drop further, as it is already very low.

### 4.1.3 Results of max score, SPECTER model

In addition to reproducing Figure 1 of the authors with the TF-IDF embeddings, we also tried to replicate the figure using the SPECTER embeddings.

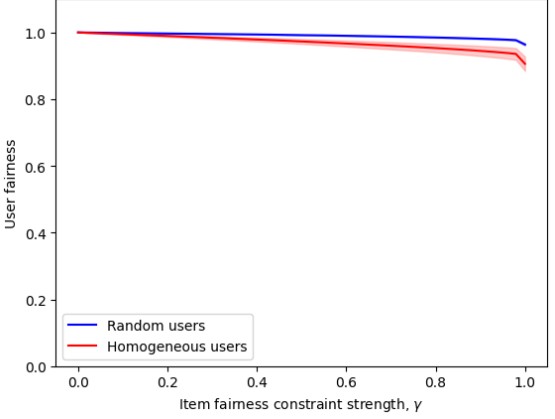
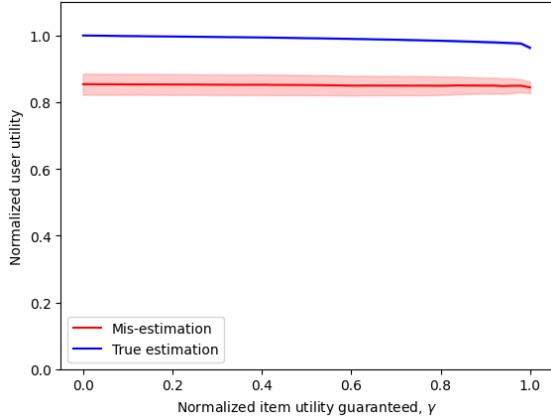

(a) Homogeneous versus diverse users.          (b) With and without misestimation.

Figure 3: Reproduction of figure 1 of Greenwood et al. (2024) using SPECTER embeddings, with a subset of the arXiv and Semantic Scholar data. (a) Shows the difference between random and homogeneous user groups when item fairness constraints are being imposed. Notable here is that, compared to the TF-IDF embeddings, the gap between random and homogeneous users is a lot smaller. (b) Shows that the price of misestimation is low, that item fairness constraints imposed on misestimated users (e.g. cold-start users) has minimal effect and that the gap between true and misestimated users is smaller than with the TF-IDF embeddings.

In Figure 3 we can see that the gap between the homogeneous and random group in Figure 3a and the gap between the true and misestimated users in Figure 3b is a lot smaller than with using the TF-IDF model. Even though the gap is smaller, the Figures still somewhat support the authors claims, as the effect of a higher item fairness constraint is worse for the misestimated group than for the random users. Another important difference with the TF-IDF model is that the price of misestimation for the misestimated group is very low. We will analyze why this is the case in the discussion (section 5).

## 4.2 Results beyond original paper

In this section, we reproduce the experiments conducted by Greenwood et al. (2024) using a different dataset to analyze whether their results are generalizable. We will again validate the recommendation engine, but this time using only *max* for the TF-IDF and SPECTER models.

### 4.2.1 Results of logistic regression, Amazon Books Reviews dataset

As can be seen in Table 4, the recommendation engine performs similarly on this dataset as it does on the original dataset. The $R^2$ is also very similar, suggesting equally predictable performance as the arXiv and Semantic Scholar data. We also conducted an extended evaluation for this dataset, which includes the same tables and figures as for the other one. The extended evaluation can be found in Appendix B. When we conducted the experiments with the Amazon Books Reviews dataset we chose to only include the *max* similarity scores for both the TF-IDF model and the SPECTER model, as the authors focus on this in the main body of their paper.

Table 4: Logistic regression results for predicting whether user $i$ reviews book $j$ highly from the similarity score $w_{ij}$ for each model. Amazon Books Reviews dataset

| Model | Coefficient | Std. Err | z-value | Adjusted $R^2$ |
|---|---|---|---|---|
| Max score, TF-IDF | 14.064204 | 0.232445 | 60.505534 | 0.422320 |
| Max score, Sentence Transformer | 48.481460 | 0.972629 | 49.845807 | 0.164829 |

### 4.2.2 Results of max score, TF-IDF model, Amazon Books Reviews dataset

Next, to verify if claims A and B generalize, we reproduce figure 1 of Greenwood et al. (2024) with the Amazon Books Reviews dataset. The reproduction using the TF-IDF embeddings can be seen in Figure 4.

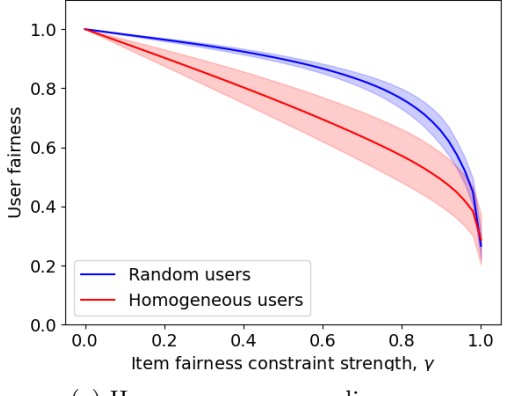

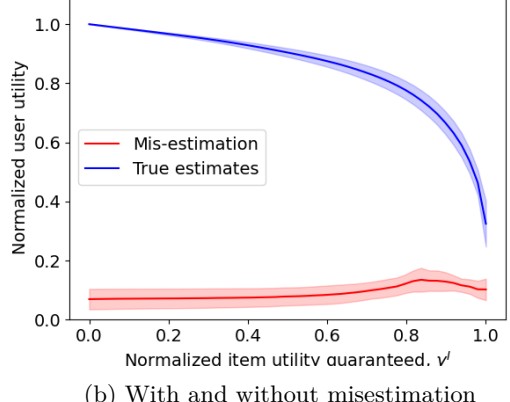

(a) Homogeneous versus diverse users.

(b) With and without misestimation

Figure 4: Reproduction of figure 1 of Greenwood et al. (2024), with the Amazon Books Reviews dataset, using TF-IDF embeddings. (a) Shows the difference between random and homogeneous user groups when item fairness constraints are being imposed. The tradeoff being better for groups with diverse preferences compared to homogeneous groups shows Claim A generalizes. (b) Shows that the cost of misestimation is high and item fairness constraints do not affect this cost, showing claim B generalizes.

The same trend as in the original figures can be observed with this dataset, suggesting that the two claims made by the original authors do generalize. Figure 4a shows that homogeneous groups are more harshly effected by higher item fairness constraints compared to groups which have diverse preferences. As $\gamma$ increases, the group which have the same preferences experience a steeper user-item fairness tradeoff. In Figure 4b we can see that the cost of misestimation for cold-start users is already so high that item fairness constraints do not affect this cost further.

### 4.3 Results of max score, SPECTER model, Amazon Books Reviews dataset

As can be seen in Figure 5, replicating Figure 1 of the authors with the SPECTER model on the Amazon Books Reviews dataset led to very similar figures as with the SPECTER model and the arXiv dataset. Once again, we can observe that the gap between the homogeneous and random groups in Figure 5a and the gap between the true and misestimated users in Figure 5b is a lot smaller than with using the TF-IDF model. However, it is worth noting that the difference between random and homogeneous users appears almost nonexistent for the Amazon Books Reviews dataset. The fact that the figures made with SPECTER embeddings are so similar for both datasets seems to support the hypothesis discussed in section 5.

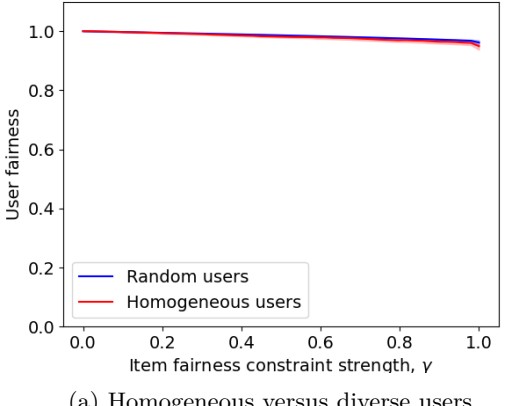 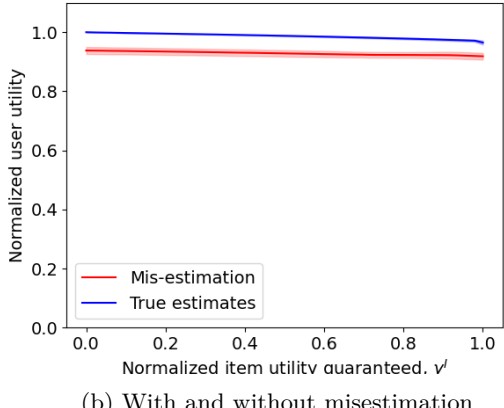

(a) Homogeneous versus diverse users.        (b) With and without misestimation

Figure 5: Reproduction of figure 1 of Greenwood et al. (2024), with the Amazon Books Reviews dataset, using SPECTER embeddings. (a) Shows the difference between random and homogeneous user groups when item fairness constraints are being imposed. As with the arXiv dataset, the difference between random and homogeneous users is not nearly as big when using SPECTER embeddings, as compared to TF-IDF. (b) Shows that the price of misestimation is low, that item fairness constraints imposed on misestimated users (e.g. cold-start users) has minimal effect and that the gap between true and misestimated users is smaller than with the TF-IDF embeddings.

## 5 Discussion

In this paper we have tried to reproduce the experiments done by Greenwood et al. (2024), to verify their two main findings. After some adjustments of the authors' code, we were able to conduct the experiments as they were done in the original work. Even though our results are not exactly the same as the ones produced by Greenwood et al. (2024), mainly due to differences in the code being used, they do completely support the authors' main claims. Figure 2a supports Claim A because, as we showed, item fairness tends to impose a higher cost on user fairness in more homogeneous populations. One difference here, however, is that user fairness for random users drops only marginally before $\gamma$ approaches its maximum value of 1. This effect is more noticeable in our results than in those of the original paper, likely due to the different sample sizes used. Figure 2b supports Claim B, as the cost of misestimation is high and item fairness constraints do not affect this cost.

Besides reproducing Figure 1 with TF-IDF embeddings, we also reproduced the figure with SPECTER embeddings. The authors focused in their paper on the TF-IDF embeddings, as they indicate in their Appendix B.4 that this is the "best performing approach" (Greenwood et al., 2024). In contradiction however, in the same section, they say that "the model using sentence transformer embeddings and the max similarity score among each author's papers appears to perform the best overall" (Greenwood et al., 2024). We believe that the authors might have chosen the TF-IDF model because it is computationally more efficient for the large scale experiments conducted, and that it aligns more with their theoretical framework than the SPECTER model. The biggest difference between the two models being that the gap between the sampled groups for both Figures is smaller for the SPECTER embeddings than for the TF-IDF embeddings. This effect can be observed with both datasets.

We believe that this difference arises primarily due to the SPECTER model being specifically trained for document-level relatedness, creating denser embeddings than the TF-IDF model. This could mean that there is less variance in the embeddings, which consequently means that the similarity scores for user-item pairs will be higher across the board. This makes it more difficult to create distinct groups for the figures; random vs. homogeneous and true vs. misestimated. This is however only speculation for now, further testing would need to be done to analyze where exactly this difference comes from.

We further extended the research in the original work by reproducing the results with the Amazon Books Reviews dataset, to determine whether their findings would generalize in a different experimental setup. With a different dataset we once again conducted the experiments. The resulting figures fully support the authors' main claims, as mentioned before. Notably, for the SPECTER embeddings, the normalized user utility under misestimated recommendations is slightly higher than that of the original dataset with the SPECTER embeddings. However, the difference is relatively small, suggesting that this is likely due to inherent differences between the datasets.

## 5.1 What was easy

The thorough and comprehensive explanation of the theoretical framework in Greenwood et al. (2024) made the paper clear and easy to follow. This was further improved by the extensive appendix, which provided additional details on the framework itself, the experimental setup, and extra experiments. Additionally, the code was publicly available, and the parts that worked were well-structured and easily understood.

## 5.2 What was difficult

The primary challenge in reproducing the authors' experiments was due to errors and missing files in the code provided by the authors. Because of this, we were unable to run the authors' original code. This suggests that the results in the paper were not generated using the provided code. This primarily hindered the creation of the data files that were required for replicating the figures. One of the missing parts was the `categories.csv` file, which they used to filter the dataset on Computer Science papers. This file was not provided by the authors so we were compelled to create our own by copying categories from the arXiv Category Taxonomy website (arXiv, n.d.).

Another part that was unclear in their code was how exactly the authors combined the arXiv dataset with Semantic Scholar data using the API. This was further complicated because we did not manage to get access to a private API key, because Semantics Scholar's API key issuance was put on pause due to high demand at the time of reproducing their work. Nevertheless, we successfully generated figures that supported both claims made in the paper.

Lastly, there were some inconsistencies in the paper, particularly in how the experiments were conducted. The paper did not clearly explain how the training and test sets were created and the code did not clarify this either, as we had to modify the code to make it functional. As a result, we had to make our own assumptions.

## 5.3 Communication with original authors

We contacted the authors with multiple questions early on, primarily regarding their provided code. For instance, we asked them whether the missing `categories.csv` file could be provided. Although the authors responded, they experienced server issues that prevented them from answering our questions within a month.

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

## A  Extended evaluation of recommendation engine

In this section we conduct the same extended evaluation of the four tested models, as done by Greenwood et al. (2024).

**Max score, TF-IDF**  Table 5 shows the average similarity scores between users and papers, based on whether a citation occurred. Figure 6 compares the similarity score distributions for cited (orange) and non-cited (blue) papers, showing higher similarity for cited papers. Table 6 presents logistic regression results predicting citation likelihood from similarity scores (Greenwood et al., 2024).

Table 5: Average similarity measures in the Max score, TF-IDF model. ArXiv and Semantic Scholar data

| Citation Type | No/Yes | Similarity Score | Score Percentile | Normalized Score |
|---|---|---|---|---|
| User cites paper | No | 0.022263 | 0.499988 | -0.000235 |
| User cites paper | Yes | 0.090412 | 0.774525 | 1.910884 |

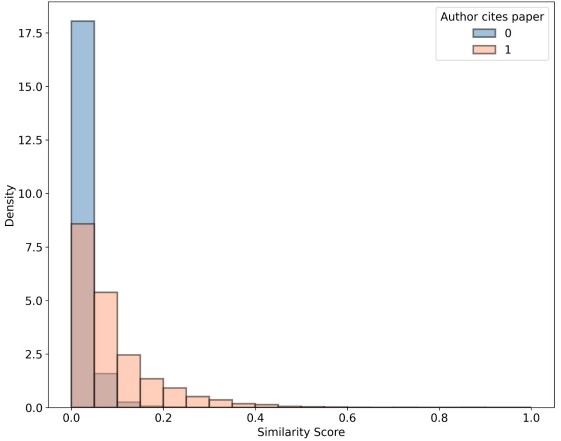

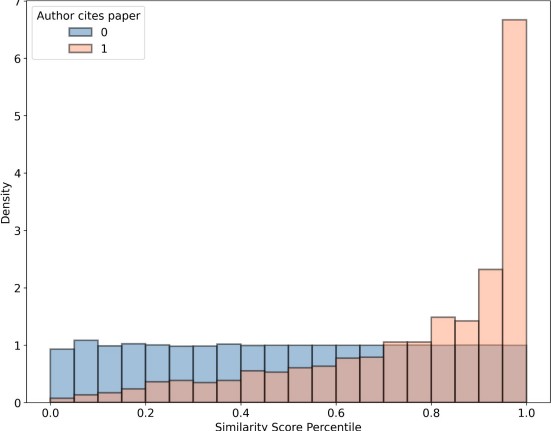

(a) Density plot of similarity scores grouped by citation presence.

(b) Density plot of similarity score percentiles grouped by citation presence.

Figure 6: Pr(score|User cites paper), the distribution of the score for a user-paper pair, conditional on whether the user cites the paper in the future, for the Max score, TF-IDF model. ArXiv and Semantic Scholar data.

Table 6: Logistic regression results for predicting whether user $i$ cites paper $j$ from the similarity score $w_{ij}$ for each model, for the Max score, TF-IDF model. ArXiv and Semantic Scholar data

| Variable | Coefficient | Std. Err | z-value | P-value | Adjusted $R^2$ |
|---|---|---|---|---|---|
| Similarity score | 15.806266 | 0.140005 | 112.897568 | 0.000000 | 0.081174 |
| Score percentile | 4.137609 | 0.085509 | 48.387914 | 0.000000 | 0.050329 |
| Normalized Score | 0.283911 | 0.003777 | 75.163223 | 0.000000 | 0.040423 |

**Mean score, TF-IDF**  Table 7 shows the average similarity scores between users and papers, based on whether a citation occurred. Figure 7 compares the similarity score distributions for cited (orange) and non-

cited (blue) papers, showing higher similarity for cited papers. Table 8 presents logistic regression results predicting citation likelihood from similarity scores (Greenwood et al., 2024).

Table 7: Average similarity measures in the Mean score, TF-IDF model. ArXiv and Semantic Scholar data

| Citation Type | No/Yes | Similarity Score | Score Percentile | Normalized Score |
|---|---|---|---|---|
| User cites paper | No | 0.017599 | 0.499986 | -0.000252 |
| User cites paper | Yes | 0.056687 | 0.787942 | 2.054151 |

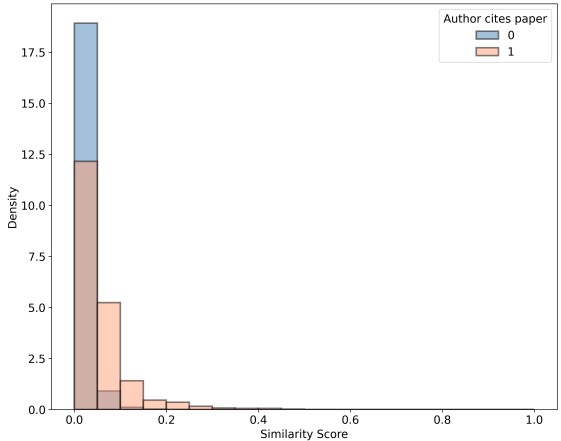

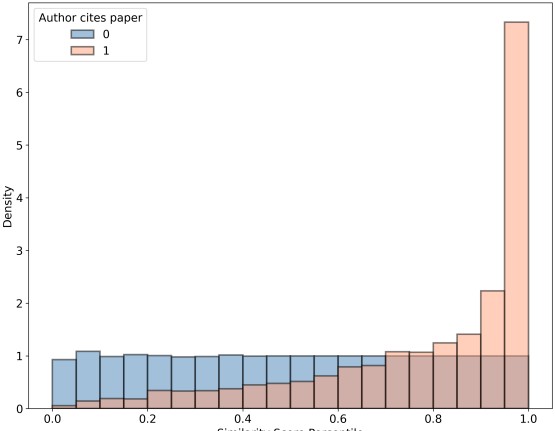

(a) Density plot of similarity scores grouped by citation presence.

(b) Density plot of similarity score percentiles grouped by citation presence.

Figure 7: Pr(score|User cites paper), the distribution of the score for a user-paper pair, conditional on whether the user cites the paper in the future, for the Mean score, TF-IDF model. ArXiv and Semantic Scholar data.

Table 8: Logistic regression results for predicting whether user $i$ cites paper $j$ from the similarity score $w_{ij}$ for each model, for the Mean score, TF-IDF model. ArXiv and Semantic Scholar data

| Variable | Coefficient | Std. Err | z-value | P-value | Adjusted $R^2$ |
|---|---|---|---|---|---|
| Similarity score | 16.960261 | 0.188324 | 90.059151 | 0.000000 | 0.050143 |
| Score percentile | 4.471924 | 0.089455 | 49.990500 | 0.000000 | 0.056098 |
| Normalized Score | 0.290417 | 0.003704 | 78.403228 | 0.000000 | 0.044561 |

**Max score, Sentence transformer** Table 9 shows the average similarity scores between users and papers, based on whether a citation occurred. Figure 8 compares the similarity score distributions for cited (orange) and non-cited (blue) papers, showing higher similarity for cited papers. Table 10 presents logistic regression results predicting citation likelihood from similarity scores (Greenwood et al., 2024).

Table 9: Average similarity measures in the Max score, Sentence Transformer model. ArXiv and Semantic Scholar data

| Citation Type | No/Yes | Similarity Score | Score Percentile | Normalized Score |
|---|---|---|---|---|
| User cites paper | No | 0.750327 | 0.499981 | -0.000161 |
| User cites paper | Yes | 0.842293 | 0.827767 | 1.307610 |

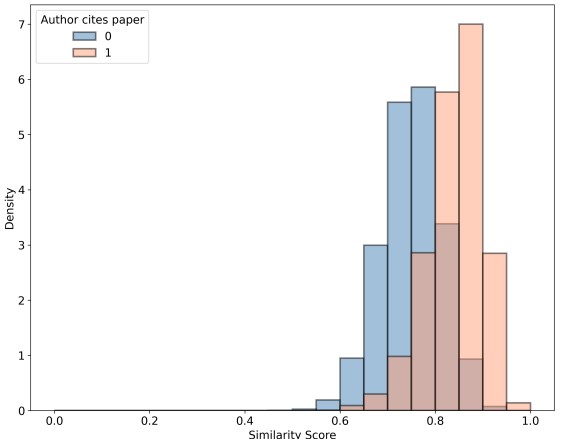

(a) Density plot of similarity scores grouped by citation presence.

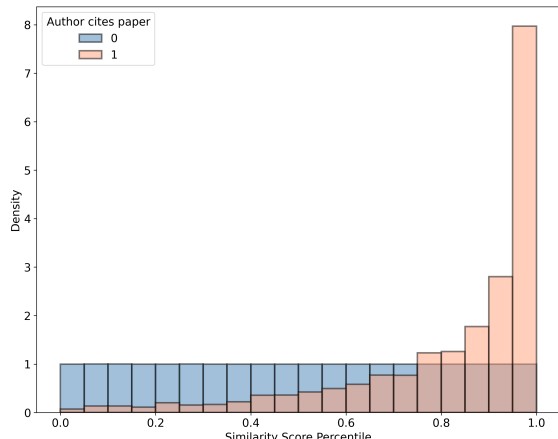

(b) Density plot of similarity score percentiles grouped by citation presence.

Figure 8: Pr(score|User cites paper), the distribution of the score for a user-paper pair, conditional on whether the user cites the paper in the future, for the Max score, Sentence Transformer model. ArXiv and Semantic Scholar data.

Table 10: Logistic regression results for predicting whether user $i$ cites paper $j$ from the similarity score $w_{ij}$ for each model, for the Max score, Sentence Transformer model. ArXiv and Semantic Scholar data

| Variable | Coefficient | Std. Err | z-value | P-value | Adjusted $R^2$ |
|---|---|---|---|---|---|
| Similarity score | 29.543296 | 0.366537 | 80.601208 | 0.000000 | 0.123819 |
| Score percentile | 5.694772 | 0.105731 | 53.861146 | 0.000000 | 0.076169 |
| Normalized Score | 1.391977 | 0.017929 | 77.638718 | 0.000000 | 0.089951 |

**Mean score, Sentence transformer** Table 11 shows the average similarity scores between users and papers, based on whether a citation occurred. Figure 9 compares the similarity score distributions for cited (orange) and non-cited (blue) papers, showing higher similarity for cited papers. Table 12 presents logistic regression results predicting citation likelihood from similarity scores (Greenwood et al., 2024).

Table 11: Average similarity measures in the Mean score, Sentence Transformer model. ArXiv and Semantic Scholar data

| Citation Type | No/Yes | Similarity Score | Score Percentile | Normalized Score |
|---|---|---|---|---|
| User cites paper | No | 0.740345 | 0.499981 | -0.000161 |
| User cites paper | Yes | 0.816375 | 0.829283 | 1.311118 |

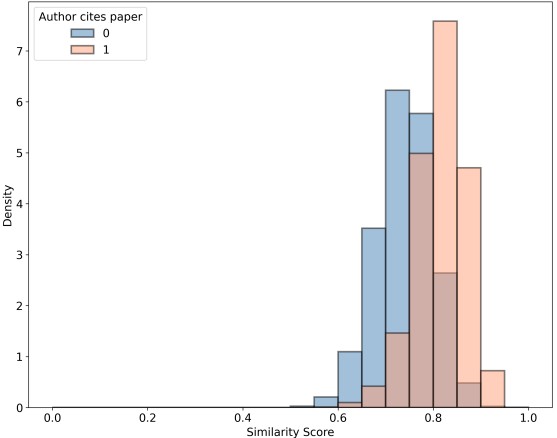

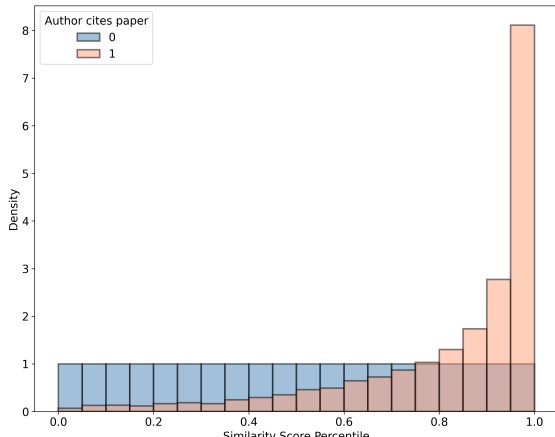

(a) Density plot of similarity scores grouped by citation presence.

(b) Density plot of similarity score percentiles grouped by citation presence.

Figure 9: Pr(score|User cites paper), the distribution of the score for a user-paper pair, conditional on whether the user cites the paper in the future, for the Mean score, Sentence Transformer model. ArXiv and Semantic Scholar data.

Table 12: Logistic regression results for predicting whether user $i$ cites paper $j$ from the similarity score $w_{ij}$ for each model, for the Mean score, Sentence Transformer model. ArXiv and Semantic Scholar data

| Variable | Coefficient | Std. Err | z-value | P-value | Adjusted $R^2$ |
|---|---|---|---|---|---|
| Similarity score | 25.713569 | 0.355772 | 72.275324 | 0.000000 | 0.091904 |
| Score percentile | 5.750422 | 0.106528 | 53.980324 | 0.000000 | 0.077036 |
| Normalized Score | 1.403841 | 0.018012 | 77.938459 | 0.000000 | 0.090813 |

# B Extended evaluation of recommendation engine, Amazon Books Reviews dataset

In this section we will conduct the extended evaluation of the Amazon Books Reviews dataset using TF-IDF and SPECTER embeddings.

**Max score, TF-IDF** Table 13 shows the average similarity scores between users and papers, based on whether a citation occurred. Figure 10 compares the similarity score distributions for cited (orange) and non-cited (blue) papers, showing higher similarity for cited papers. Table 14 presents logistic regression results predicting citation likelihood from similarity scores (Greenwood et al., 2024).

Table 13: Average similarity measures in the Max score, TF-IDF model. Amazon Books Reviews dataset

| Citation Type | No/Yes | Similarity Score | Score Percentile | Normalized Score |
|---|---|---|---|---|
| User reviews book highly | No | 0.031760 | 0.500041 | -0.014166 |
| User reviews book highly | Yes | 0.447842 | 0.851719 | 7.880599 |

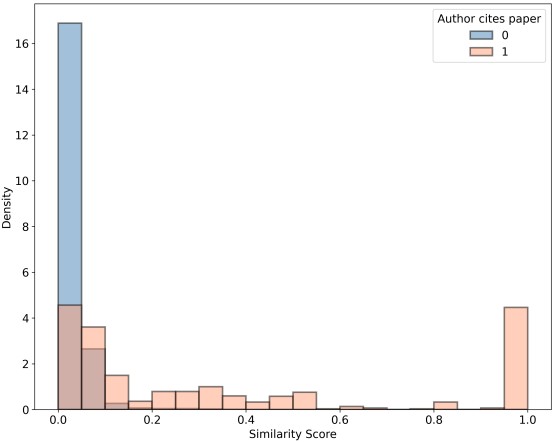

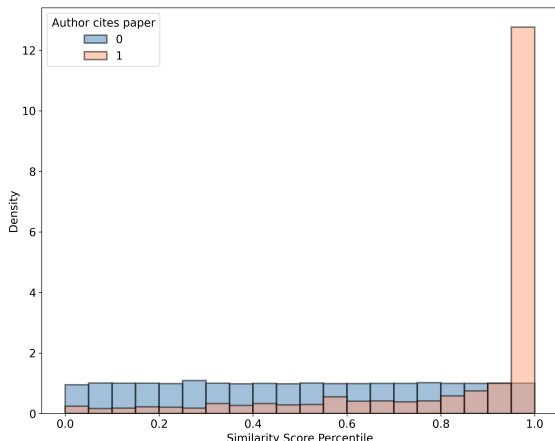

(a) Density plot of similarity scores grouped by high review.

(b) Density plot of similarity score percentiles grouped by high review.

Figure 10: Pr(score|User highly rates book), the distribution of the score for a user-review pair, conditional on whether the user rates the book a four or higher, for the Max score, TF-IDF model. Amazon Books Reviews dataset.

Table 14: Logistic regression results for predicting whether user $i$ highly rates book $j$ from the similarity score $w_{ij}$ for each model, for the Max score, TF-IDF model. Amazon Books Reviews dataset

| Variable | Coefficient | Std. Err | z-value | Adjusted $R^2$ |
|---|---|---|---|---|
| Similarity score | 14.064204 | 0.232445 | 60.505534 | 0.422320 |
| Score percentile | 6.672986 | 0.187991 | 35.496373 | 0.124009 |
| Normalized Score | 0.553874 | 0.007183 | 77.106529 | 0.376168 |

**Max score, Sentence transformer** Table 15 shows the average similarity scores between users and papers, based on whether a citation occurred. Figure 11 compares the similarity score distributions for cited (orange) and non-cited (blue) papers, showing higher similarity for cited papers. Table 16 presents logistic regression results predicting citation likelihood from similarity scores (Greenwood et al., 2024).

Table 15: Average similarity measures in the Max score, SPECTER model. Amazon Books Reviews dataset

| Citation Type | No/Yes | Similarity Score | Score Percentile | Normalized Score |
|---|---|---|---|---|
| User reviews book highly | No | 0.857845 | 0.500204 | -0.001922 |
| User reviews book highly | Yes | 0.927471 | 0.752460 | 1.034501 |

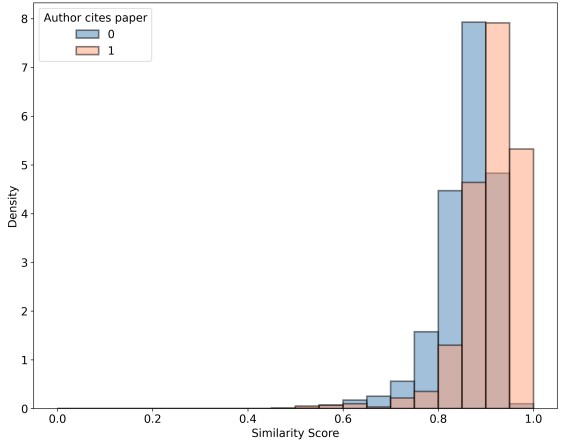

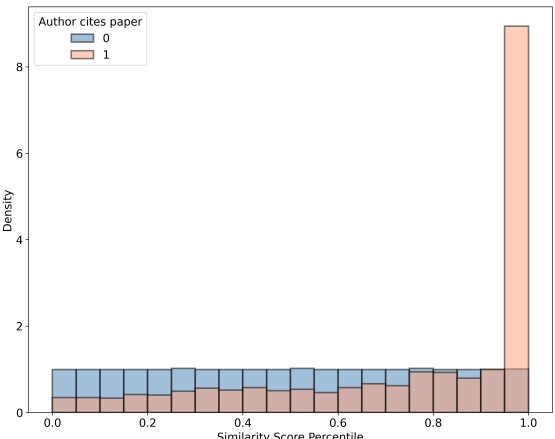

(a) Density plot of similarity scores grouped by high review.

(b) Density plot of similarity score percentiles grouped by high review.

Figure 11: Pr(score|User highly rates book), the distribution of the score for a user-review pair, conditional on whether the user rates the book a four or higher, for the Max score, SPECTER model. Amazon Books Reviews dataset.

Table 16: Logistic regression results for predicting whether user $i$ highly rates book $j$ from the similarity score $w_{ij}$ for each model, for the Max score, SPECTER model. Amazon Books Reviews dataset

| Variable | Coefficient | Std. Err | z-value | Adjusted $R^2$ |
|---|---|---|---|---|
| Similarity score | 50.970792 | 0.955893 | 53.322705 | 0.182179 |
| Score percentile | 3.638931 | 0.123186 | 29.540122 | 0.057065 |
| Normalized Score | 2.360930 | 0.046494 | 50.779515 | 0.139233 |

## C Extended evaluation of recommendation engine, using the same data ratio

To see whether keeping the same ratio (1/10 test/train) as the authors makes a substantial difference, we will reproduce the results here with keeping this ratio in tact. This means that we limit our testing data to 1/10 of a fraction of our train data. The experiments conducted below are therefore executed with a train size of $82,737$ and test size of $8,497$. We will only report the max score of the TF-IDF model, as this is the main method used by the authors. As we found no meaningful differences when keeping the same ratio or not, we decided to use all our available data.

**Max score, TF-IDF**  Table 17 shows the average similarity scores between users and papers, based on whether a citation occurred. Figure 12 compares the similarity score distributions for cited (orange) and non-cited (blue) papers, showing higher similarity for cited papers. Table 18 presents logistic regression results predicting citation likelihood from similarity scores (Greenwood et al., 2024).

Table 17: Average similarity measures in the Max score, TF-IDF model. ArXiv and Semantic Scholar data

| Citation Type | No/Yes | Similarity Score | Score Percentile | Normalized Score |
|---|---|---|---|---|
| User cites paper | No | 0.022284 | 0.500026 | -0.000225 |
| User cites paper | Yes | 0.088162 | 0.770517 | 1.843296 |

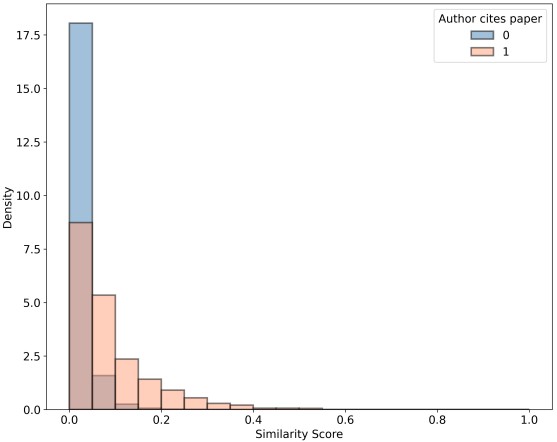

(a) Density plot of similarity scores grouped by citation presence.

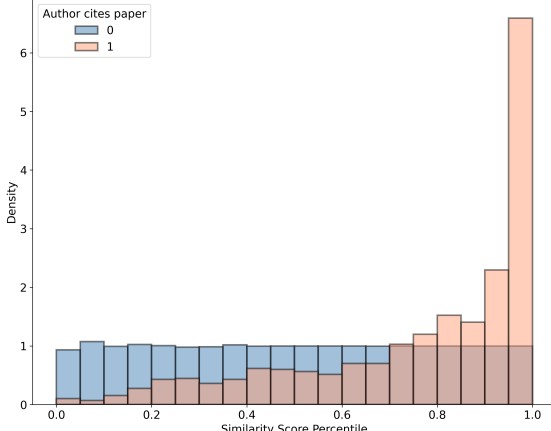

(b) Density plot of similarity score percentiles grouped by citation presence.

Figure 12: Pr(score|User cites paper), the distribution of the score for a user-paper pair, conditional on whether the user cites the paper in the future, for the Max score, TF-IDF model. ArXiv and Semantic Scholar data.

Table 18: Logistic regression results for predicting whether user $i$ cites paper $j$ from the similarity score $w_{ij}$ for each model, for the Max score, TF-IDF model. ArXiv and Semantic Scholar data

| Variable | Coefficient | Std. Err | z-value | P-value | Adjusted $R^2$ |
|----------|-------------|----------|---------|---------|----------------|
| Similarity score | 15.793567 | 0.234398 | 67.379262 | 0.000000 | 0.078444 |
| Score percentile | 4.042440 | 0.141031 | 28.663444 | 0.000000 | 0.048639 |
| Normalized Score | 0.290717 | 0.006294 | 46.189846 | 0.000000 | 0.040192 |

# D    Distribution of categories in dataset

To assess the differences in sizes between our train and test sets and those of the authors, we plotted the distribution of subcategories within the Computer Science category and compared them with the plots in Greenwood et al. (2024). There are some differences that can be observed. For instance, while the highest peak in both the train and test sets of the authors is the subcategory 'Computer Vision and Pattern Recognition', in our case, the highest peak corresponds to the 'Machine Learning' subcategory. Another significant difference is the sheer size of the subcategories, the authors' have bigger subcategories for the train dataset, and we have bigger subcategories for the test dataset. One notable observation here is that the authors' test dataset distribution does not seem to add up to their specified $14,307$ papers, as the highest subcategory only contains 600 papers. The subcategories however generally follow the same trend, with the most prominent ones being consistent across both our plots and those of the authors. We can therefore assume that this will not affect the experiments conducted in the paper.

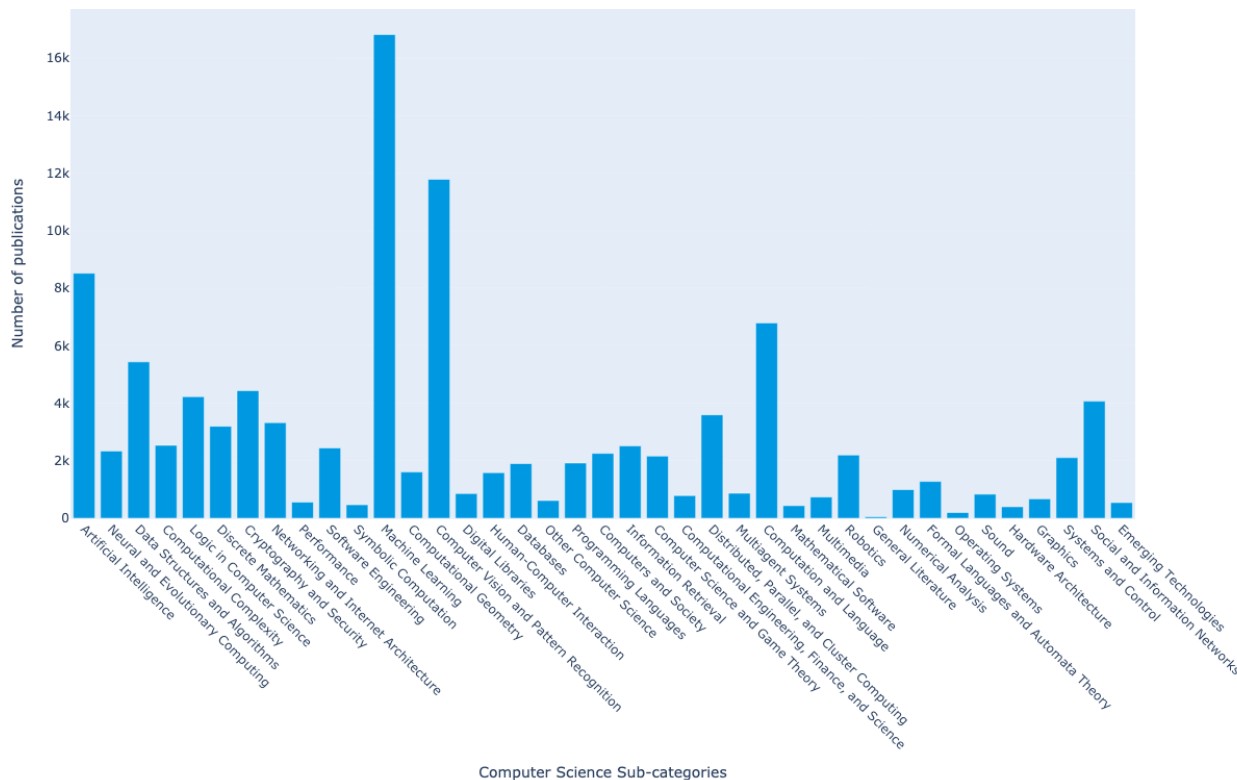

(a) Distribution of research paper publications in the train dataset over time, for this paper.

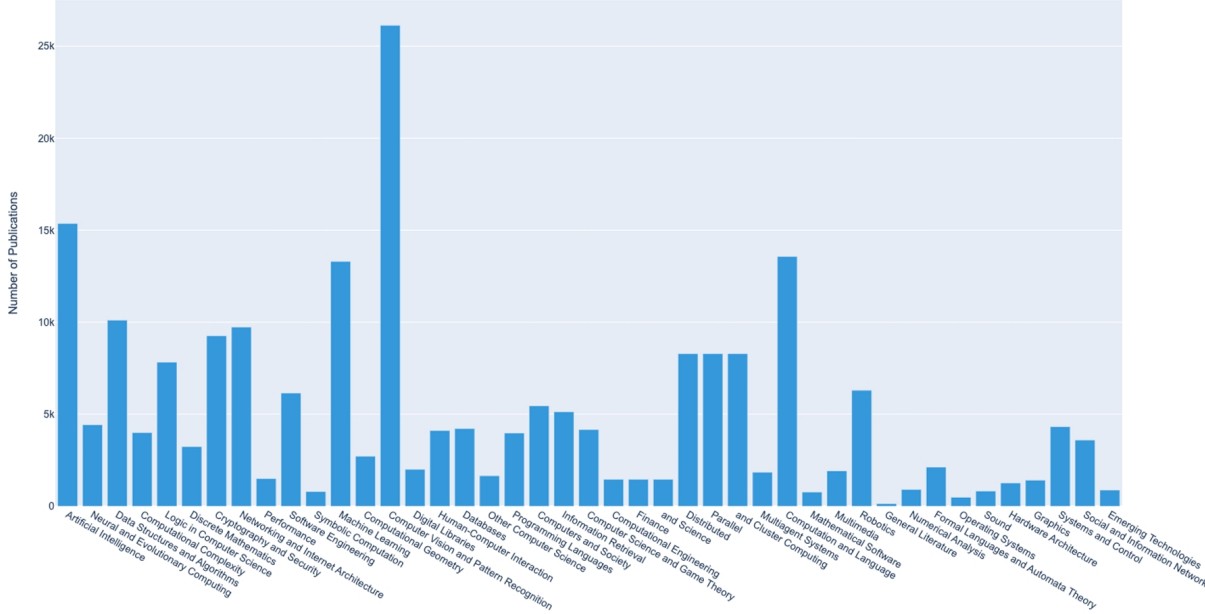

(b) Distribution of research paper publications in the train dataset over time, used in the original paper.

Figure 13: Comparison between distribution of the original and our train dataset over the different Computer Science Sub-categories. ArXiv and Semantic Scholar data.

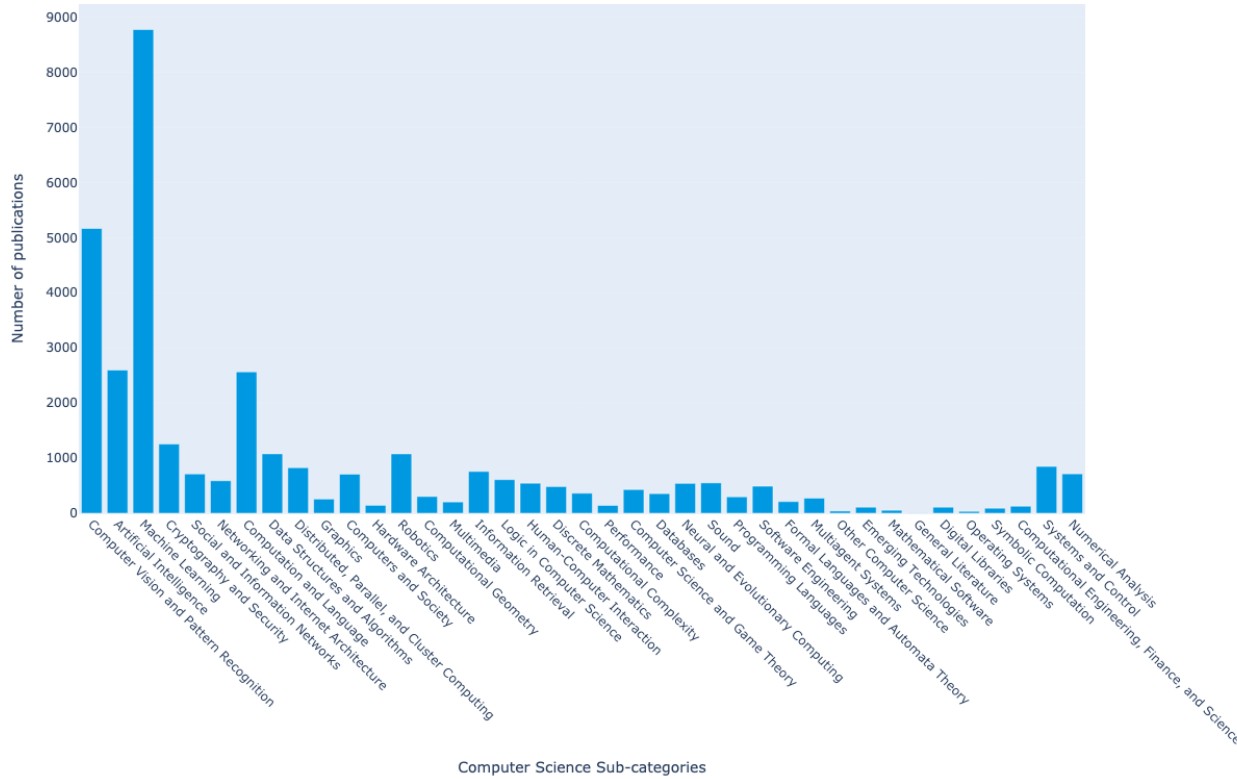

(a) Distribution of research paper publications in the test dataset over time, used in this paper.

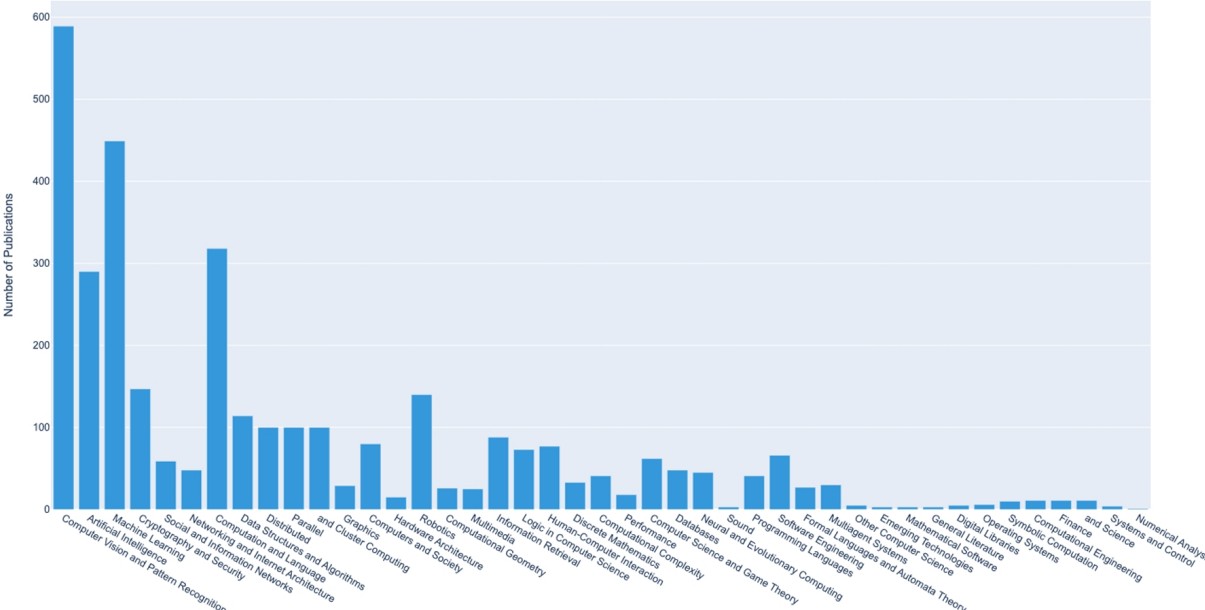

(b) Distribution of research paper publications in the test dataset over time, used in the original paper.

Figure 14: Comparison between distribution of the original and our test dataset over the different Computer Science Sub-categories. ArXiv and Semantic Scholar data.

