# OpenReview forum: "Beyond TF-IDF: Reproducibility and Generalizability of Two-sided Fairness Tradeoffs"
_TMLR — Withdrawn by Authors_

### Review · Reviewer_u7nF · 2025-03-02

**Summary Of Contributions:**

This paper aims at reproducing and validating the experimental results from Greenwood et al. (2024). The major contributions of this manuscript are:
- Successful reproduction of the main empirical results.
- Extension to the experiments via a new Amazon Book dataset to validate the results of the original paper .

**Audience:**

No

**Broader Impact Concerns:**

This paper does not discuss broader impact concerns. However, since the topic is related to fairness, I believe it should contribute to both the benefit and fairness of the ML community.

**Claims And Evidence:**

No

**Requested Changes:**

Regarding generalisable insights:
- The authors should imrpove their work on extracting broader insights on how two-sided fairness tradeoffs behave across different datasets or embedding methods, rather than simply confirming prior results. Currently, there is only one more dataset to validate, and lacks of explanations of the new results, especially for the SPECTER model (see weakness)

Regarding actionable lessons:
- The authors could provide takeaways for practitioners, such as conditions under which SPECTER embeddings might be preferable and when TF-IDF might be a better choice when using the proposed method.
- In the original paper, Greenwood et al. attempted to extend the method to other fairness definitions, so it would be valuable to provide clear guidance on when and how different fairness constraints should be applied in real-world recommendation systems.
- The proposed method currently supports recommending only a single item, this limitation should be discussed since in practice, this is not always the case, the real-world recommendation systems typically suggest multiple items.

Others:
- The whole Section 5 looks weird. The discussion section should address the limitations of the current work, how the experimental results connect with the theoretical aspects, and the planned improvements for future work, rather than personal reflections.

**Strengths And Weaknesses:**

Strengths:
- Except from using the dataset used in the original paper (arXiv and Semantic Scholar), this paper extends the original experiments by using a new dataset (Amazon Book), which broadening the scope of the case study.
- Table 3 is closely matches Table 1 from the original paper, which helps validate the robustness of the methods.

Weakness:
- The motivation for selecting Greenwood et al. (2024) in the introduction is not very clear. At least, the authors should explain the main contribution of the original paper to highlight the significance of the problem being addressed, as well as its limitations, which justify the need for validation. For example, the original paper provides a theoretical framework, but its findings have not been validated for robustness and generalizability, which is critical for assessing whether the proposed method is applicable in real-world settings.
- The explanation of the results on the new dataset is weak. The authors state, “However, it is worth noting that the difference between random and homogeneous users appears almost nonexistent for the Amazon Books Reviews dataset,” but provide limited insights (which is shown in section 5) for this observation.
- “Even though the gap is smaller, the Figures still somewhat support the authors’ claims, as the effect of a higher item fairness constraint is worse for the misestimated group than for the random users.” is vague and lacks rigor. Is there any statistical evidence supporting the claim that the figures using SPECTER still align with the original authors’ conclusions?
- In Figure 5(a), the results for random users and homogeneous users are almost identical. This seems questionable, were the results produced correctly? Is there any explanation for this?
- No clear reasoning is given for why the gap using the SPECTER model is much smaller than when using TF-IDF. I checked Section 5 for an explanation of the SPECTER model’s behavior, but the discussion is weak and lacks supporting evidence. The authors argue that SPECTER’s denser embeddings reduce variance and obscure group distinctions, yet they did not test this hypothesis on datasets optimized for SPECTER, such as citation-based corpora. Why not?

Here are some questions:
- In Section 4.1.2, “We therefore chose to reproduce the figure with a subset of the data,” what was the selection criterion for the subset? How were the reported results defined? Since the gap in Figure 2(a) is much smaller than in the original paper, could this be due to how the subset was selected?
- In Figure 3, why is the standard deviation for random users in (a) and misestimation in (b) almost zero? (Same question for Figure 5.)

---

### Review · Reviewer_dQgR · 2025-03-17

**Summary Of Contributions:**

The authors perform a replication study of a recent two-sided fairness paper by Greenwood et al (2024). They aim to investigate whether two main phenomena / takeaways of that paper regarding the tradeoffs associated with two-sided fair recommendation systems hold water empirically:

(1) Interaction with estimation quality: For users whose preferences have not been adequately estimated for whatever reason, fairness constraints may have outsized negative impact.

(2) Interaction with preference homogeneity: For groups of users with homogeneous preferences intra-group, they can be especially impacted by item fairness constraints (e.g. as everyone's top 1 item preference may be the same and therefore swapping out for any other item would disadvantage everyone) --- and conversely, for users with diverse preferences, fairness may come for free without having to necessarily be enforced.

The authors replicate the experiments on Arxiv data from the previous paper, using TF-idf and SPECTER to embed textual data. They additionally re-run the experiments on another dataset, Amazon Books, to confirm the findings. Despite the (publicly available) code for the previous paper being incomplete/underspecified in several ways, they are able to qualitatively confirm the paper's findings on their "native" experiments as well as on the Amazon Books data, for TF-IDF embedding. On the new observations front, they then obtain that the observations of Greenwood et al (2024) are only present in a very mild form if at all for SPECTER embedding on both datasets: namely, the sign of the effects (of homogeneity and misestimation) identified by Greenwood et al is still correct but its magnitude can be vanishing.

**Audience:**

No

**Broader Impact Concerns:**

N/A.

**Claims And Evidence:**

Yes

**Requested Changes:**

To sum up my opinion, the current study doesn't sufficiently expand the horizons of Greenwood et al at the current moment. Especially given the very modest amount of experimentation done in the source paper, this study needs to venture boldly into other settings. Please see above for a list of possible improvements to the study. They come in the form of proposals to augment this reproducibility study with extra experiments from a variety of angles and criteria that have not been considered by this study (and by the original paper). Admittedly, starting from its current state this study would need to undergo some significant expansion to satisfy these criteria. Most importantly, besides pointing out that it appears that some embeddings may lead to diminished magnitudes of effects identified by Greenwood et al, I believe the TMLR audience would need more substantiation for that phenomenon --- as well as at least several more generalizable new insights into the problem, giving the audience insights into how/whether the conclusions of Greenwood et al hold across a variety of settings/tasks within and beyond text-based recommendations.

**Strengths And Weaknesses:**

The strength of the paper is that (1) it is overall well-written and that (2) it transparently reproduces and verifies the findings of the original paper as well as (3) contributes a new insight --- that the effects identified by Greenwood et al can sometimes be barely observable.

That being said, I believe that in its current state, this reproducibility study falls substantially short on the side of both quantity and quality of insights. Overall, the original paper Greenwood et al (2024) operated within a very broad two-sided fairness framework theoretically speaking, and therefore to me, having read that paper while reviewing this study, the standout direction for reproducing and robustifying that paper's empirical insights/claims would be to experiment with a broad range of settings/tasks involving users and items with preferences. Here are just some example directions that come to mind:

--- The study identifies an interesting observation: seemingly, with more powerful/tailored embedding like SPECTER, which can possibly be much better/low-variance at helping estimate true user preferences, the studied effects lose their magnitude. But it doesn't follow through on that observation: For instance a natural question is, it this diminishing effect phenomenon true more broadly simply as a function of how good the embedding is? For this, this study should probably be rerun on other good embeddings than just SPECTER: For instance, Google's Universal Sentence Encoder; or some more modern and powerful embeddings: e.g. OpenAI text-embedding-3; or some other encodings/embeddings. For each of these embeddings, it would then be pertinent to study what estimated preference structure they lead to: homogeneous? heterogeneous? well-estimated? etc.

--- The user-item system doesn't have to be text-based; in fact, getting rid of the necessity to perform textual embeddings would be a great way to ablate away from having to factor in the SPECTER vs tf-idf difference. Greenwood et al's framework extends very broadly. So natural candidates such as MovieLens and other datasets in non-textual domains come to mind.

--- If one insists to stay within text-based user/item settings, there are at least several other datasets besides Amazon Books that can be studied in similar ways: For instance, the Goodreads Book Graphs datasets collected by UCSD (https://cseweb.ucsd.edu/~jmcauley/datasets/goodreads.html) come to mind.

--- There are some interesting findings/questions that were already apparent in the Greenwood et al paper, and confirmed by the current study but largely left on the table.

--- For one example, what about the user-item fairness tradeoff curve being largely flat except for extreme values of the item fairness parameter gamma? What causes this? Can one find data on which the tradeoff is more steep?

--- What about other aggregation rules than mean/max over the users preferences? And what about recommending more than 1 item?

--- What about settings where users exhibit multiple types (within each type, the preferences would be homogeneous) --- as the Greenwood et al paper proposed for future work to investigate?

--- It would be good to construct various synthetic datasets investigating the above open questions in a more controlled fashion: E.g. the user types question could be studied in detail; and various definitions of "misestimation", other than just the one in Greenwood et al, could be more comprehensively studied.

---

### Review · Reviewer_k7Nk · 2025-03-17

**Summary Of Contributions:**

The authors examine the problem of two-sided fairness in recommendation systems by replicating the experiments conducted by Greenwood et al. (2024). Their study aims to assess the reproducibility of the original findings and provide further validation of the proposed fairness tradeoffs. In addition to faithfully reproducing the original experiments, the authors extend the analysis by incorporating a different dataset, allowing them to evaluate the generalizability of Greenwood et al.'s claims.

**Audience:**

No

**Claims And Evidence:**

No

**Requested Changes:**

Please refer to the Weaknesses part.

**Strengths And Weaknesses:**

**Strengths**:

1. The authors successfully reproduced the experiments conducted by Greenwood et al. (2024) despite facing limitations due to insufficient data and incomplete source code.

2. There are no obvious typos in the paper writing.

**Major Weaknesses**:


1. The paper lacks a clearly articulated motivation, making it difficult to understand the significance and objectives of the study.

2. While the authors present performance results, they do not offer new insights beyond those already established by Greenwood et al. (2024) regarding the two-sided fairness problem.

3. The authors compare the performance of SPECTER and TF-IDF embeddings but do not conduct additional experiments to empirically support their analysis.

4. The abstract states that the study extends the original experiments by incorporating SPECTER embeddings and the Amazon Books dataset to assess generalizability. However, Greenwood et al. (2024) already included SPECTER embedding results in their original work.

5. The dataset used in Section 3.4.1 was significantly downscaled to 1,430 papers and 2,051 authors, which is much smaller than the dataset used by Greenwood et al. (2024). This reduction in scale could introduce substantial randomness and impact the reliability of the results.

6. The authors employ the Amazon Books Reviews dataset and consider a recommendation to be successful if a user rates a book above 4 stars. However, this approach may not be well-suited for analyzing two-sided fairness, as users who are uninterested in a recommended book may simply ignore it rather than leave a review, leading to potential biases.

7. The claim that "The similar $R^2$ scores suggest that our model has comparable predictive performance to the one used by the authors" is not well-founded, since the authors used a different train/test dataset from Greenwood et al. (2024). Moreover, $R^2$ scores alone are insufficient to demonstrate model similarity, as they only capture one aspect of predictive performance.

8. In Section 4.2.1, the authors chose to report only the Max similarity scores for both the TF-IDF and SPECTER models. However, Greenwood et al. (2024) provided results for both Max and Mean similarity scores in their main paper and appendix. To ensure a more comprehensive analysis, the authors should consider including both.

**Minor Weaknesses**:

1. In Section 1, the authors define user fairness as "a measure of the relevance of the recommended articles." Since the paper does not exclusively focus on article recommendation, it would be more appropriate to replace "articles" with "items."

2. The Amazon Books Reviews dataset is used without proper citation. The authors should include a reference to ensure proper attribution.

---

### Author Response · Authors · 2025-03-31
**Reaction to the reviews**

Dear reviewers, thank you for your careful consideration of our reproducibility study! We appreciate the recommendations and directions for extending our paper that you gave. However, we have realized that  we do not have enough time to resolve some of our paper's major weaknesses. So we have decided to withdraw our submission. Again, thank you for you detailed reviews. It is very valuable to us, as we can take this feedback with us to consider in future projects.

---

### Note · Authors · 2025-04-11

**Comment:**

Dear reviewers, thank you for your careful consideration of our reproducibility study! We appreciate the recommendations and directions for extending our paper that you gave. However, we have realized that we do not have enough time to resolve some of our paper's major weaknesses. So we have decided to withdraw our submission. Again, thank you for you detailed reviews. It is very valuable to us, as we can take this feedback with us to consider in future projects.

**Withdrawal Confirmation:**

I have read and agree with the venue's withdrawal policy on behalf of myself and my co-authors.